# Disability and the achievement of Universal Health Coverage in the Maldives

**Lena Morgon Banks**[1][�]*, **Timothy O'Fallon**[1][�], **Shaffa Hameed**[1], **Sofoora Kawsar Usman**[2], **Sarah Polack**[1], **Hannah Kuper**[1]

1 International Centre for Evidence in Disability, London School of Hygiene & Tropical Medicine, London, United Kingdom, 2 Es-Key, Maldives

he These authors contributed equally to this work.
* morgon.banks@lshtm.ac.uk

## Abstract

### Objective

To assess access to general and disability-related health care among people with disabilities in the Maldives.

### Methods

This study uses data from a case-control study (n = 711) nested within a population-based, nationally representative survey to compare health status and access to general healthcare amongst people with and without disabilities. Cases and controls were matched by gender, location and age. Unmet need for disability-related healthcare is also assessed. Multivariate regression was used for comparisons between people with and without disabilities.

### Results

People with disabilities had poorer levels of health compared to people without disabilities, including poorer self-rated health, increased likelihood of having a chronic condition and of having had a serious health event in the previous 12 months. Although most people with and without disabilities sought care when needed, people with disabilities were much more likely to report difficulties when routinely accessing healthcare services compared to people without disabilities. Additionally, 24% of people with disabilities reported an unmet need for disability-related healthcare, which was highest amongst people with hearing, communication and cognitive difficulties, as well as amongst older adults and people living in the lowest income per capita quartile. Median healthcare spending in the past month was modest for people with and without disabilities. However, people with disabilities appear to have high episodic healthcare costs, such as for disability-related healthcare and when experiencing a serious health event.

### Conclusions

This study found evidence that people with disabilities experience unmet needs for both disability-related and general healthcare. There is therefore evidence that people with

**Data Availability Statement:** The underlying data can be found on LSHTM Data Compass (https://datacompass.lshtm.ac.uk/id/eprint/1698/). It is available by request in which the user is asked to fill in a short form in order to ascertain that they will

be using the data for the purpose of reproduction or other legitimate scientific purposes. The decision to grant access is not determined by any of the authors, but rather independently by the public repository. Access on request is to prevent the data from being used for purposes other than replication or other scientific inquiry. The restrictions were put in place at the request of the data protection office at the grant holding institution. Individuals are able to access the data in the same manner as the authors and the authors do not have any special privileges to access the data.

**Funding:** This research was funded by 3ie (Grant number: PW3.11.MDV.IE; Grant holder: HK; https://www.3ieimpact.org/) and the UK National Institute for Health and Care Research (grant: 100273IR10; Grant holder: HK). The funders had no role in study design, data collection and analysis, decision to publish, or preparation of the manuscript.

**Competing interests:** The authors have declared that no competing interests exist.

disabilities in the Maldives are falling behind in core components relevant to UHC: availability of all services needed, and quality and affordability of healthcare.

## Introduction

Strengthening health systems and addressing inequalities in access to healthcare to ensure healthy lives for all is a core Sustainable Development Goal (Goal 3) [1]. This goal includes a target to achieve Universal Health Coverage (UHC), meaning that there is access to quality healthcare for all, including the full range of services needed, with financial protection. Here too there is a focus on left behind groups, including people with disabilities. However, few studies have considered UHC from a disability perspective [2].

There is a strong rationale for a focus on people with disabilities in the journey towards UHC. They are a large group, making up one billion people, or one in seven people globally [3]. Furthermore, on average, people with disabilities have worse health than their peers without disabilities resulting from their underlying health condition/impairment and a higher prevalence of both proximal (e.g. obesity, lack of exercise, poor diet) and distal (e.g. poverty, discrimination, poor living conditions) risk factors [2, 3]. People with disabilities will therefore have greater general healthcare needs, on average. They also often require disability-related healthcare, including rehabilitation and assistive devices [4].

Evidence suggests, however, that people with disabilities are being left behind by health systems despite their greater need. They often experience greater challenges in accessing healthcare services, due to informational, financial, physical and attitudinal barriers [2]. They are also affected by large service gaps, particularly for rehabilitation [5]. Poor availability, quality and affordability of services can lead to unmet needs for many people with disabilities, resulting in worsening health and functioning [2]. These gaps may be particularly pronounced for people with disabilities living in rural areas, or with certain types of impairments [4, 6, 7]. They may also incur higher healthcare costs, as they may need to seek care more often and incur additional costs in doing so (e.g. paying for a companion to travel with them, accessible transport), yet have a lower capacity to pay due to their overrepresentation amongst people living in poverty [8]. In many settings, people with disabilities are more likely to experience catastrophic health expenditures compared to people without disabilities [3, 9, 10]. For example, the World Report on Disability found that across 51 countries, half of people with disabilities could not afford needed health services and people with disabilities were more than 50% more likely to report catastrophic health expenditures compared to people without disabilities [3].

Inclusive health systems are therefore important in the attainment of UHC. The right to health and healthcare for people with disabilities is also supported by international law, including the United Nations' Convention on the Rights of Persons with Disabilities, and the laws of most countries [11]. However, policies and programmes supporting these rights are often lacking or not put into practice or enforced. This neglect is reinforced by gaps in evidence documenting whether and how people with disabilities are deprived of health and healthcare, particularly in low- and middle-income countries (LMICs). Consequently, this study aims to explore access to general and disability-related health care among people with disabilities in the Maldives. This will be assessed using data from a nationally-representative, population-based survey with a nested case-control study [12]. It will cover health needs and access to both general and disability-related health services.

## Study context

The Maldives is an upper-middle income country in South Asia that has made impressive development and public health gains in the past decades [13, 14]. A third of the population lives in the capital Malé, with the remainder dispersed across 186 inhabited islands [13]. Approximately 7% of Maldivian citizens have a disability [12].

In 2018, the Government of the Maldives spent 9% of gross domestic product (GDP) on health, far higher than other countries in the region (regional average is 3.5% of GDP) [15]. *Husnuvaa aasandha ("Aasandha")* is the national social health insurance scheme, which covers all Maldivian citizens and is financed through the national budget [16]. It was enacted in 2011 through the National Social Health Insurance Act and is run by the National Social Protection Agency (NSPA) operating under the Ministry of Health [17]. *Aasandha* covers inpatient and outpatient care, costs for medications and for transport in emergency cases, with no caps on spending for eligible services [16]. Services are free at point of use, although contributions are required if accessing services in the private sector, or without appropriate referrals [18]. The Maldives also runs Medical Welfare that can provide coverage of services, devices and treatments not covered through *Aasandha* (e.g. care in private hospitals, medical devices such as oxygen machines) [19]. Applications are made on an ad hoc basis and require an application to NSPA.

The Maldives has a four-tier healthcare system. Each inhabited island has a health centre from which patients can be referred to higher levels facilities in the atolls, region and central (Malé) level [20]. Given the geographic dispersion of the population, travel can be a significant cost in accessing healthcare particularly for people living outside Malé [21]. Further, travel abroad for healthcare is common in the Maldives, as some services are not available or perceived to be of inferior quality [21, 22]. *Aasandha* covers the travel and direct medical costs for some overseas medical healthcare from contracted providers in India and Sri Lanka, if the services are not available in the Maldives (e.g. certain types of cancer treatment), and the individual receives a referral from a public sector specialist doctor [22, 23]. Still, one study estimated that Maldivians spent on average US$204 per capita in 2013 on overseas medical travel [23].

The Disability Act (2010) codifies the rights of people with disabilities to equal access to healthcare. *Aasandha* covers most disability-related health services (e.g. rehabilitation, psychiatry, ophthalmology), although many services are heavily centralised or not available in the country, necessitating domestic and overseas travel. Assistive devices, and their repair and replacement, are not covered through *Aasandha*, but can be provided through Medical Welfare [19].

## Methods

This paper uses data from a nationally-representative population-based survey of disability and a nested case-control study comparing age-sex-location matched people with and without disabilities.

## Sample selection

Data collection took place between July to August 2017. Participants were identified through a nationally-representative population-based survey and nested case-control study, the methods for which has been described in detail elsewhere [12]. In brief, a target sample size of 6,500 people aged 2+ was set and participants were selected through a two-stage sampling strategy (probability proportionate to size, followed by compact segment sampling).

All enumerated individuals were screened for disability using the Washington Group Short Set Enhanced for adults (18+) and the UNICEF-Washington Group Child Functioning

Modules (sets for children 2–4, 5–17) [24–26]. These questions focus on difficulties with every-day activities (e.g. seeing, hearing, walking, remembering) and most have four response options on level of difficulty experienced in performing each activity (none, some, a lot, cannot do). People were defined as having a disability if they reported "a lot of difficulty" or "cannot do at all" for at least one question or experienced "daily" symptoms of anxiety or depression at an intensity described as "a lot" (adults 18+; "daily" symptoms for children 5–17). People were also defined as having a disability if they received the Disability Allowance or reported a health condition that made them eligible for the Disability Allowance (e.g. psychosocial impairments such as schizophrenia, bipolar disorder; autism spectrum disorder) [27].

All people identified as having a disability were invited to participate in the nested case-control study. Each "case" with a disability was matched to a "control" without a disability, who was also drawn from the same population-based survey. Matching variables were age (+/- 5 years), gender and location (same survey cluster, or else same administrative island/atoll). Controls could not be from households with other members with a disability. Controls were selected at random if multiple eligible controls were available for a case. All participants were Maldivian citizens and thus eligible for *Aasandha* and Medical Welfare.

## Selection of health indicators

Data was collected from both cases and controls on their health status and access to general healthcare. This included questions on: diagnosis of and treatment for chronic conditions; EQ-5D self-reported rating of current health on a scale from 0 to 100 (with 0 being worst imaginable state of health and 100 the best imaginable state of health) [28]; occurrence of a serious health event in the last 12 months and experience accessing care; frequency of experiencing different challenges when accessing healthcare (tool from Demographic and Health Surveys [29]); if they had private health insurance outside of *Aasandha* or had ever received Medical Welfare; and household healthcare spending in the last month. Healthcare expenditures were captured for the following: 1) household healthcare expenditures in the last month–which covered direct costs for general and disability-related healthcare services and products–and were presented as total per capita, as a percentage of household income and if they qualified as catastrophic (25% or more of household income) [30]; 2) costs associated with seeking care amongst people who had a serious health problem in the last 12 months, including direct (i.e. costs for services) and indirect (i.e. costs for accommodations, travel) expenditures; 3) lifetime spending on disability-related healthcare services and products.

People with disabilities were asked about their access to disability-related healthcare (e.g. rehabilitation, assistive devices) during the population-based survey. This included questions on: if they had ever been to health professional about the difficulties they faced for each functional domain; their awareness of impairment-relevant services/devices (e.g. glasses/ophthalmology for people with difficulty seeing); their perceived need and use of a service/device. Reporting needing, but not using a service/device, was categorised as self-reported 'unmet need'. Coverage was calculated as the proportion who reported using a service out of those who reported a need (i.e. use/need). If people experienced multiple functional limitations, they were asked about each limitation separately. Respondents were also asked about their total spending on disability-related healthcare.

## Data analysis

Data was analysed using STATA 16. Indicators on general health and healthcare access were compared between cases and controls. For each variable on general health and healthcare access (e.g. health-related quality of life, having private health insurance), a multivariate

regression was run, which included variables for age, gender and location (Malé vs other atolls) as well as disability status and the outcome of interest. Met and unmet needs for disability-related services were compared amongst people with disabilities by functional domain. Further, sociodemographic and economic predictors of unmet need for disability-related services were assessed through logistic regression, adjusting for the individual's age, gender, and location.

For healthcare costs, medians and interquartile ranges (IQR) were compared using a Mann-Whitney test.

## Ethical considerations

Ethical approval was granted by the ethics board at the London School of Hygiene & Tropical Medicine in the United Kingdom, the Maldives National Bureau of Statistics and the Ministry of Health's National Health Research Council. All study protocols, including for consent, were approved by these bodies.

Written or audio recorded consent was obtained for all study participants. Audio consent was used for interviews conducted by phone (e.g. household members temporarily working on other islands, fishing). In these instances, the full consent process was audio recorded and saved. Cases and controls were interviewed directly where possible, with healthcare expenditures answered by the household member who was most knowledgeable of household finances. Carer consent was sought for minors (<18 years) and people with impairments that severely limited their ability to communicate/understand, and assent received from the individual if they were able to self-report on any of the questions.

## Results

Overall, 5,363 people aged 2+ were screened for disability (response rate: 82%) in the population-based survey, of whom 403 were identified as having a disability.

The case-control study included 380 cases and 331 controls (response rate: 90.1%). Cases and controls were similar in age and gender, although cases were more likely to live in Malé compared to controls and were in households with lower per capita incomes (Table 1).

## Access to general healthcare

Overall, people with disabilities had poorer health status than people without disabilities (Table 2). People with disabilities were more likely to have experienced a health problem in the last 12 months (19% vs 8% for controls, aOR = 2.5, 95%CI: 1.5, 4.0), and to have been diagnosed with at least one chronic condition (46% vs 33% for people without disabilities, p = 0.001). People with disabilities reported significantly lower health ratings using the WHO-QOL-BREF tool (average score 55.5 vs 72.6 for people without disabilities, p<0.001).

Healthcare coverage was similar amongst people with and without disabilities, whether measured in terms of seeking treatment for a health problem or for a specific chronic condition (Table 3). However, people with disabilities were more likely to report difficulties when accessing health services compared to people without disabilities. When asked about their typical experience accessing health services, people with disabilities were significantly more likely to report experiencing difficulties "often" for almost all questions, with the exception of lacking female service providers among women. For example, in comparison to people without disabilities, people with disabilities were about twice as likely to report often having difficulties with negative attitudes from staff and the distance and transport to facilities compared to people without disabilities and almost three times as likely to have difficulties paying for services and in getting someone to accompany them when seeking services. People with disabilities

**Table 1. Description of the study sample (case-control).**

| | Cases (n = 380) | Controls (n = 331) | OR (95% CI)* |
|---|---|---|---|
| *Gender* | | | |
| Male | 163 (43%) | 135 (41%) | Reference |
| Female | 217 (57%) | 196 (59%) | 0.9 (0.7, 1.2) |
| *Age group* | | | |
| 2–17 | 61 (16%) | 48 (15%) | Reference |
| 18–39 | 94 (25%) | 87 (26%) | 0.9 (0.5, 1.4) |
| 40–64 | 123 (32%) | 125 (38%) | 0.8 (0.5, 1.2) |
| 65+ | 102 (27%) | 71 (22%) | 1.1 (0.7, 1.8) |
| *Location* | | | |
| Malé | 151 (40%) | 85 (26%) | Reference |
| Other atolls | 229 (60%) | 246 (74%) | 1.9 (1.4, 2.6) |
| *Functional domain[†]* | | | |
| Vision | 88 (23%) | | |
| Hearing | 35 (9%) | | |
| Physical | 190 (49%) | | |
| Mental health | 81 (21%) | | |
| Cognitive | 75 (20%) | | |
| Communication | 56 (15%) | | |
| Median annual per capita household income (SD) | $2,625 ($4,384) | $3045 ($5,361) | p = 0.004 |

*Bivariate analysis

[†]Categories are not mutually exclusive as some participants reported multiple disabilities.

were less likely to report being satisfied with the services they received compared to people without disabilities, although this difference was not statistically significant.

## Access to disability-related healthcare

Overall, 76% of people with disabilities reported seeing a healthcare professional for their functional difficulties (Table 4). By functional domain, health seeking was lowest for cognition (53%) and mental health (50%), and highest for physical (92%) and vision (90%) impairments. Most cases had heard of a specific service/device across each functional domain (>80%), while this was lower for cognition (61%). Self-reported need (reporting that a service/device would

**Table 2. Health status amongst people with and without disabilities.**

| | Disability (n, %) | No disability (n, %) | aOR (95% CI) |
|---|---|---|---|
| Experienced a health problem in last 12 months | 71 (19%) | 27 (8%) | 2.5 (1.5, 4.0)*** |
| *Diagnosed with a chronic condition[1]* | | | |
| • Diabetes | 53 (17%) | 33 (12%) | 1.4 (0.9, 2.3) |
| • Hypertension | 108 (34%) | 59 (21%) | 2.0 (1.3, 3.1)** |
| • Asthma | 36 (11%) | 23 (8%) | 1.4 (0.8, 2.4) |
| • Any of above | 147 (46%) | 88 (31%) | 1.9 (1.3–2.7)** |
| | Mean (SD) | Mean (SD) | |
| Health rating (0–100) [1] | 51.5 (29.0) | 72.9 (22.0) | p<0.001*** |

* p < .05

** p < .01 (adjusted for age, gender, location)

[1] Amongst people aged 18+

**Table 3. Access to general health services amongst people with and without disabilities.**

| | Disability (n, %) | No disability (n, %) | aOR (95% CI) |
|---|---|---|---|
| *Current treatment for chronic conditions*[§] | | | |
| Diabetes | 43 (81%) | 29 (88%) | 0.6 (0.1, 3.1) |
| Hypertension | 95 (88%) | 52 (88%) | 1.4 (0.5, 4.1) |
| Asthma | 22 (61%) | 10 (43%) | 2.2 (0.7, 7.4) |
| Any of above[‡] | 113 (77%) | 66 (75%) | 1.3 (0.7–2.4) |
| *Health seeking amongst people experiencing a health problem in last 12 months*[ß] | | | |
| Sought treatment for health problem | 68 (96%) | 27 (100%) | n/a |
| *Where went for services*[ß] | | | |
| Same island | 25 (37%) | 8 (30%) | Reference |
| Another island | 32 (47%) | 13 (48%) | 1.1 (0.3, 3.8) |
| Abroad | 11 (16%) | 6 (22%) | 0.7 (0.2, 2.8) |
| At least somewhat satisfied with services received[ß] | 54 (79%) | 25 (93%) | 0.3 (0.1, 1.5) |
| *Reported difficulties when typically using health services*[†] | | | |
| Paying for services | 119 (31.5%) | 39 (11.8%) | 3.7 (2.4, 5.5)*** |
| Distance to facility | 148 (39.4%) | 66 (20.1%) | 2.9 (2.1, 4.2)*** |
| Transport to facility | 165 (43.9%) | 79 (24.0%) | 2.7 (1.9, 3.8)*** |
| Having someone accompany | 91 (24.2%) | 28 (8.5%) | 3.7 (2.3, 5.8)*** |
| (Women only) Lack of female service providers | 43 (20.0%) | 51 (26.2%) | 0.8 (0.5, 1.2) |
| Staff availability | 156 (41.3%) | 110 (33.5%) | 1.6 (1.1, 2.2)** |
| Lack of medications | 166 (43.9%) | 106 (32.1%) | 1.8 (1.3, 2.5)*** |
| Negative attitudes from staff | 55 (14.6%) | 27 (8.2%) | 2.0 (1.2, 3.4)** |

[§]Amongst people reporting being diagnosed with a chronic condition

[‡] If reported multiple chronic conditions, treatment coverage defined as having received treatment for all conditions

[ß] Amongst those who experienced a health problem in the last 12 months and sought treatment

* p < .05

** p < .01, p<0.001 (adjusted for age, gender, location)

[†]Reported experiencing difficulties 'often' compared to 'never'/'sometimes'

be helpful) was higher for physical (71%), vision (67%) and communication (59%) domains and lower for hearing, mental health and cognitive (<50%). However, reported use (current or ever) of a service/device for their self-perceived functional limitations was generally low across all domains; 53% of participants reporting physical difficulties had used a service/device

**Table 4. Met and unmet needs for disability-related healthcare among people with disabilities by functional domain.**

| | Vision | Hearing | Physical | Mental Health | Cognitive | Communication | Overall |
|---|---|---|---|---|---|---|---|
| | N = 88 | N = 35 | N = 190 | N = 81 | N = 75 | N = 56 | N = 385 |
| | N (%) | N (%) | N (%) | N (%) | N (%) | N (%) | N(%) |
| Ever been to a doctor about difficulties* | 79 (90%) | 27 (77%) | 174 (92%) | 41 (50%) | 40 (53%) | 42 (75%) | 293 (76%) |
| Aware of service/device for difficulties* | 81 (92%) | 28 (80%) | 175 (92%) | 65 (80%) | 46 (61%) | 49 (88%) | 307 (80%) |
| Expressed need for service/ device* | 59 (67%) | 16 (46%) | 135 (71%) | 32 (40%) | 36 (48%) | 33 (59%) | 229 (60%) |
| Use of service/ device* | 35 (40%) | 4 (11%) | 101 (53%) | 28 (35%) | 15 (20%) | 13 (23%) | 161 (42%) |
| Unmet need for service/device** | 24 (27%) | 12 (34%) | 34 (18%) | 4 (5%) | 21 (28%) | 20 (36%) | 94 (24%) |
| Coverage** | 59% | 25% | 75% | 88% | 42% | 39% | 79% |

*Denominator is the total number of people reporting any limitation in that functional domain and thus categories are not mutually exclusive; **Coverage is calculated as use/need

**Table 5. Healthcare spending.**

| | Disability | No disability | Measure of association |
|---|---|---|---|
| *Household spending on healthcare, previous month* | | | |
| Total (median, IQR) | $0.65 ($8.70) | $0.34 ($9.32) | p = 0.51 |
| Total as a proportion of household income (median, IQR) | 0.4% (5%) | 0.2% (4%) | p = 0.24 |
| Catastrophic healthcare expenditures[1] (n, %) | 45 (13%) | 38 (10%) | aOR = 0.8 (95%CI: 0.5, 1.3) |
| *Healthcare spending when seeking care for a serious health problem, previous year[2]* | | | |
| Direct costs (median, IQR) | $93.85 ($970.87) | $194.17 ($2543.69) | p = 0.80 |
| Indirect costs (median, IQR) | $177.99 ($811.42) | $388.35 ($1035.60) | p = 0.17 |
| Overall costs (median, IQR) | $501.62 ($2297.74) | $828.48 ($3216.83) | p = 0.25 |
| *Use of social health protection products* | | | |
| Has private health insurance (n, %) | 21 (6%) | 26 (8%) | aOR = 0.6 (95%CI: 0.3, 1.2) |
| Has ever received Medical Welfare (n, %) | 18 (5%) | 3 (1%) | aOR = 6.8 (95%CI: 1.9, 23.5)** |
| *Spending on disability-related healthcare (lifetime)* | | | |
| Total (median, IQR) | $1,100.32 ($3,883.50) | n/a | n/a |

[1] Healthcare spending 25% or more of household income

* p < .05

** p < .01 (adjusted for age, gender, location); [2]Amongst people experiencing a health problem in the last 12 months, n = 68 for cases, n = 27 controls; Rate of exchange used: 1 USD = 14.45 MVR

and 40% for those with a vision limitation. This was even lower for the domains of mental health (35%), communication (23%), cognition (20%) and hearing (11%). Self-reported unmet need for services/device was highest for the domains of communication (36%) and hearing (34%) and lowest for mental health (5%). Overall, coverage (use/perceived need) was high for the mental health (88%) and physical domains (75%) and lowest for communication (39%) and hearing (34%). Reported need increased significantly with age (S1 Table). Unmet need increased significantly with age and was more common among people living in the lowest, compared to the highest, income per capita quartile (S2 Table). No other demographic or economic variables were significantly associated with reported need or unmet need.

## Healthcare financing

Median per capita expenditure in the last month was generally low, at $0.65 (IQR = $8.70) for people with disabilities and $0.34 (IQR = $9.32) for people without disabilities, which equated to a median of 0.4% and 0.2% of household income (Table 5). However, there was high variability in healthcare spending, with 13.4% of people with disabilities and 10.2% of people without disabilities living in household experiencing catastrophic health expenditures in the last month (greater than 25% of household income). Health spending was high when seeking services for a serious health problem in the past year; people with and without disabilities reported spending a median of $502 and $828 respectively (p = ns). Few people had private health insurance or had received Medical Welfare, although people with disabilities were significantly more likely to have received Medical Welfare compared to people without disabilities (4.8% vs 0.9%, aOR = 6.8, 95% CI: 1.9, 23.5).

People with disabilities reported high and variable costs for disability-related healthcare, with a median total spending of $1,100.32 (IQR: $3,883.50).

## Discussion

This study found evidence that people with disabilities experience unmet needs for both disability-related and general healthcare. For general health, people with disabilities had poorer

levels of health compared to people without disabilities, including poorer self-rated health, increased likelihood of having a chronic condition and of having had a serious health event in the previous 12 months. Although most people with and without disabilities sought care when needed, people with disabilities were much more likely to report difficulties when routinely accessing healthcare services compared to people without disabilities. Additionally, 24% of people with disabilities reported an unmet need for disability-related healthcare, which was highest amongst people with hearing, communication and cognitive difficulties, as well as amongst older adults and people living in the lowest income per capita quartile. Median healthcare spending in the past month was modest for people with and without disabilities. However, people with disabilities appear to have high episodic healthcare costs, such as for disability-related healthcare and when experiencing a serious health event. There is therefore evidence that people with disabilities in the Maldives are falling behind in core components relevant to UHC: availability of all services needed, and quality and affordability of healthcare.

Our finding that people with disabilities have poorer health status, on average, is consistent with the wider literature. For example, as in this study, people with disabilities were more likely to report a serious health event in the last months compared to people with disabilities in studies in India, Cameroon, Vietnam and Nepal [31, 32]. The more frequent reporting of specific chronic health conditions among people with disabilities, including diabetes, is also mirrored in other studies [33], including in LMICs such as Guatemala [34], Malawi [35], and South Africa [36].

In contrast, there was little difference in treatment coverage–for chronic conditions or when seeking urgent care–between people with and without disabilities in the Maldives. This finding is in contrast to other literature from LMICs [37]. Potentially, the availability of *Aasandha* in the Maldives, which covers most healthcare services, contributed to good healthcare coverage among people with disabilities. Furthermore, knowledge of and access to disability-related health services were generally higher than in other studies in LMICs such as Bangladesh, India, Cameroon and Haiti (Danquah et al., 2015; Mactaggart et al., 2015; Pryor et al., 2018). For example, a study in Bangladesh found 70% of people with disabilities had an unmet need for an assistive product [38]. Again, the availability of *Aasandha* as well as Medical Welfare (which cover other services not included in *Aasandha*, such as assistive devices) may have supported the high coverage. Still, only 5% of people with disabilities had ever accessed Medical Welfare (for any reason) and spending on disability-related healthcare was high (median: $1,100.32, IQR: $3,883.50). Further research is required to understand the low use of Medical Welfare amongst people with disabilities, particularly as it is the main source of provision for assistive devices. Interventions such awareness campaigns, providing support with applications, or refining eligibility criteria/the application process should be explored to increase use of this programme amongst people with disabilities.

People with disabilities in the Maldives reported difficulties in accessing healthcare services and with quality of care, which is consistent with the available literature, particularly from LMICs [37, 39]. Other studies have reported much higher out-of-pocket healthcare spending than this study amongst people with disabilities, and greater unmet needs compared to people without disabilities [40, 41]. For example, in Vietnam 30% of people with disabilities covered through social health insurance still faced catastrophic health expenditure spending, which was significantly more compared to other insured groups [42]. The relatively low spending on healthcare in the Maldives may reflect the strengths of the national health insurance programme *Aasandha*, which does not have individual spending caps and covers many disability-related health services–which is in contrast to many other health financing schemes in LMICs [43, 44]. Alternatively, healthcare expenditures may have been high but infrequent, and so not adequately captured in the one month recall period. There is evidence to support this theory,

as people with disabilities had high lifetime costs for disability-related healthcare (median: $1,100.32, IQR: $3,883.50). Further, costs for seeking treatment amongst the 19% of people with disabilities who had a health problem in the previous year were high (median $501, IQR: $2,297). Indirect costs (e.g. for travel) were a significant source of spending, and were not captured in the monthly recall period for household healthcare spending. High indirect and opportunity costs associated with seeking healthcare have been found in other studies [45–47]. These costs are particularly high in the Maldives–and other low population density and/or island nations—due to the lack of economies of scale needed to support the provision of tertiary care (e.g. many disability-related services) [22]. Consequently, many people must travel from remote islands to the capital Malé or abroad to receive needed care [22, 23]. Further research is needed in the Maldives and other settings on the role of health insurance and other programmes in improving access to healthcare and reducing out-of-pocket direct and indirect costs for people with disabilities.

The generally good coverage of disability-related services conceals variation by sub-group, as low household wealth, age and functional domain were predictors of unmet need for disability-related services. Other research has found cost to be a key factor in poor access to disability-related services, along with poor availability and centralisation of services [38, 48–50], which may explain higher unmet need amongst people living in poverty. Similarly, difference in coverage by functional domain may be linked to the geographic spread of services. For example, coverage for mental health, physical and vision were relatively high (88%, 59% and 75%, respectively), while coverage for people with hearing and communication limitations was low (25% and 39% respectively). This variation could reflect the availability and location of services in the Maldives. While healthcare services for physical and visual impairments are available at most atoll regional health centres, hearing services are very limited and only available in Malé. Alternatively, people may not be aware of what services would be beneficial to them. For example, mental health coverage was high (88%), mainly because few (40%) reported needing services. Studies from other settings have found that people may not recognise mental health conditions as treatable health conditions, be aware of services that could improve their symptoms or that self-stigmatisation may inhibit individuals from recognising the need for services [51–53].

## Limitations

In interpreting the results of this study, several considerations should be taken into account. Assessments of household healthcare only included direct costs and spending within the last month. Some healthcare expenditures, such as for disability-related services, may be high but infrequent and thus not captured fully within this recall period for a study of this size. Additionally, some unmet health needs for both disability-related and general healthcare may not have been captured as they were based on self-report. Capturing unmet health needs is methodologically challenging, as individuals may not know about products and services that could improve their functioning, particularly if awareness of these items is limited or requires specialist assessment [54]. Further, the survey only captured if an individual who reported needing disability-related health services/products was using it, but did not measure whether they were sufficient to meet the individual's needs. Consequently, assessments of unmet need are likely underestimated. Another concern is that this study focused on Maldivian citizens (84% of the population [55]). However, non-citizens, such as migrant workers–who are not entitled to *Aasandha* but must purchase health insurance as a condition of their work visas– likely have different experiences accessing healthcare. Further research is also required to explore in greater detail differences in access to general health services by characteristics such

as gender, impairment type and location. In terms of strengths, the study was relatively large, nationally representative, and included comprehensive measures of disability and healthcare access.

## Conclusion

Health discrepancies for people with disabilities may be less pronounced in the Maldives than in other LMICs, potentially because of the existence of a comprehensive national health insurance programme. Such initiatives may therefore not only support progression to UHC, but also ensure that people with disabilities are not left behind. The health system in the Maldives has several elements of good practice that could guide other countries in developing disability-inclusive UHC. Importantly, *Aasandha* provides wide coverage for most general and disability-related health services, is free at the point of use and does not have spending caps for eligible services. Medical Welfare is also available for services not covered on *Aasandha*, such as assistive devices.

Still, there are areas for improvement to ensure the Maldives is better able to meet its commitments to UHC for people with disabilities, which also carry implications for other settings. Importantly, centralisation of services and travel are major barriers to accessing required services, particularly for disability-related services as most are located in the capital Malé. Further, episodic costs appear high for both urgent care and disability-related service. Additional social protection programmes, or wider uptake of Medical Welfare, may help to offset some of the indirect costs of seeking required care. Similarly, decentralising services such as through community-based or outreach programmes could improve affordability and availability.

## Supporting information

**S1 Table. Predictors of reporting a need for an assistive device among people with disabilities.**
(DOCX)

**S2 Table. Predictors of reporting an unmet need for an assistive device or specialist service among people with disabilities.**
(DOCX)

**S1 File.**
(DOCX)

## Acknowledgments

Electronic data solutions were provided by LSHTM Global Health Analytics (odk.lshtm.ac. uk).

## Author Contributions

**Conceptualization:** Lena Morgon Banks, Timothy O'Fallon, Hannah Kuper.

**Data curation:** Lena Morgon Banks.

**Formal analysis:** Lena Morgon Banks, Timothy O'Fallon, Sarah Polack.

**Funding acquisition:** Hannah Kuper.

**Investigation:** Lena Morgon Banks, Shaffa Hameed, Sofoora Kawsar Usman.

**Methodology:** Lena Morgon Banks, Sarah Polack, Hannah Kuper.

**Project administration:** Lena Morgon Banks, Shaffa Hameed, Sofoora Kawsar Usman.

**Supervision:** Shaffa Hameed, Sarah Polack, Hannah Kuper.

**Writing – original draft:** Lena Morgon Banks.

**Writing – review & editing:** Timothy O'Fallon, Shaffa Hameed, Sofoora Kawsar Usman, Sarah Polack, Hannah Kuper.

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
