## [Decision Letter · Decision Letter 0]

14 Mar 2022

PONE-D-21-33864Disability and the achievement of Universal Health Coverage in the MaldivesPLOS ONE

Dear Dr. Banks,

Thank you for submitting your manuscript to PLOS ONE. After careful consideration, we feel that it has merit but does not fully meet PLOS ONE’s publication criteria as it currently stands. Therefore, we invite you to submit a revised version of the manuscript that addresses the points raised during the review process. Both reviewers have by and large positively evaluated the rigour and scope of the study. However, they also outline (in particualr reviewer #1) that there is scope for improving the manuscript further.. Several clarifications are required in order to help the reader appreciate the relevance of the study. I invite you to follow closely all the suggestions included in the reports and to change the manuscript accordingly before resubmitting the paper. I also ecourage you to read carefully PLOS ONE data policy and to ensure that your submission meets all the requirements.

We look forward to receiving your revised manuscript.

Kind regards,

Matteo Lippi Bruni, PhD

Academic Editor

PLOS ONE

Journal Requirements:

2. In the ethics statement in the Methods, you have specified that verbal consent was obtained. Please provide additional details regarding how this consent was documented and witnessed, and state whether this was approved by the IRB. In addition, in your Methods section, please ensure you have also stated whether you obtained consent from parents or guardians of the minors included in the study or whether the research ethics committee or IRB specifically waived the need for their consent.

3. Please include a complete copy of PLOS’ questionnaire on inclusivity in global research in your revised manuscript. Our policy for research in this area aims to improve transparency in the reporting of research performed outside of researchers’ own country or community. The policy applies to researchers who have travelled to a different country to conduct research, research with Indigenous populations or their lands, and research on cultural artefacts. The questionnaire can also be requested at the journal’s discretion for any other submissions, even if these conditions are not met.  Please find more information on the policy and a link to download a blank copy of the questionnaire here: https://journals.plos.org/plosone/s/best-practices-in-research-reporting. Please upload a completed version of your questionnaire as Supporting Information when you resubmit your manuscript.

Reviewers' comments:

Reviewer's Responses to Questions

**Comments to the Author**

1. Is the manuscript technically sound, and do the data support the conclusions?

Reviewer #1: Yes

Reviewer #2: Yes

2. Has the statistical analysis been performed appropriately and rigorously? 

Reviewer #1: Yes

Reviewer #2: Yes

3. Have the authors made all data underlying the findings in their manuscript fully available?

Reviewer #1: Yes

Reviewer #2: No

4. Is the manuscript presented in an intelligible fashion and written in standard English?

Reviewer #1: Yes

Reviewer #2: Yes

5. Review Comments to the Author

Reviewer #1: INTRODUCTION

1. Clarify the difference between unmet needs and expected restrictions by nature of disability

2. Explain briefly how COVID-19 has impacted on service delivery and access specific to disability given insurance, that is the differences before and after COVID-19 that were observed

3. Briefly highlight the evidence in literature in addition to saying its there, explain it in relation to the findings/study context, include some figures

4. Highlight the policies on disability that are not being enforced or lacking that will preclude recommendations later on

5. In introduction give a systematic regional and global picture for context that will be used as a preamble to the explanations in the discussion

STUDY CONTEXT

1. Highlight the problem with access and disability clearly as Maldives has made strides and accommodates various health problems via insurance

2. Explain the problem using means such as OOP, CHE. Consider reporting mortality or hospital admissions due to chronic illness as opposed to merely reporting a chronic health condition

3. Consider adding more exclusion criteria such as severe disability

METHODS AND RESULTS

1. The model may be overfitted and may be difficult to replicate. Having too many variables can cause multicollinearity for example, expenditure in last month and expenditure in last 12 months overlap, consider using one of the two

2. Define what a serious health event is for the problem to be highlighted

3. Consider mentioning how variables affect health status, QALYs for example for readers to appreciate the problem more

4. The variable self health rating needs further clarification as one may see it as something any individual with a chronic illness would rate as poor

5. The majority of people with disability in the study sample are aged 40-64+. This is an interesting finding as it may justify the good healthcare system in Maldives and brings in the possibility of occupational hazards being poorly managed and the need for geriatric care to include disability.

6. Compare episodic costs for disabled and non-disabled to appreciate the high cost, or compare the episodic costs to those experienced by disabled people in countries in the region

DISCUSSION

1. Explain the policy or benefits package deficiencies in relation to results for example more effort should be placed in registration, is the study advocating for specific benefits packages for the disabled?

2. Explain all statements empirically for example good coverage conceals variation in coverage by sub-groups

3. When comparing results from other studies, highlight the differences or similarities to your study and the meanings they bring

4. When explaining or stating results, state the significance of the findings in relation to UHC, and recommendations

5. Consider categorizing variables or findings into knowledge and perception, expenditure/costs, service provision, health status, access (distance, transport) for a systematic flow of the discussion

6. Revise some grammatical and referencing errors

Reviewer #2: It is an article that provides important and relevant evidence about access to healthcare services for persons with disabilities.

Comments

LMIC the name is not described before the acronyum

The methodology should be clear about which variables were used in the logistic regression,

In children younger than 18 which questions were used to assess disability?

It will be important to present the average household income of persons with and without disabilities in the sample.

It will be recommended to include in the methodology which variables were used to adjust the logistic regresion (apart from sex and age), also to explain what happened to the control variables and to include interactions, such as sex and disability and analyse if the coefficients changes.

In addition, to include clear policy recommendations in the context of Maldives and to explain why the Maldives is a good country to study this topic, and how other countries can learn from these findings.

6. PLOS authors have the option to publish the peer review history of their article (what does this mean?). If published, this will include your full peer review and any attached files.

Reviewer #1: **Yes: **Agness Ngwira

Reviewer #2: No

---

## [Author Response · Author response to Decision Letter 0]

10 Sep 2022

August 15, 2022

Dear PLoS One Editors,

We thank the two reviewers and the editor for their comments, which we feel have strengthened this paper. Below we have detailed our responses to each reviewer comment. 

Best

Morgon Banks

Reviewer 1

1. Clarify the difference between unmet needs and expected restrictions by nature of disability

We have added more detail to the third paragraph of the introduction.

2. Explain briefly how COVID-19 has impacted on service delivery and access specific to disability given insurance, that is the differences before and after COVID-19 that were observed

This research was collected before the onset of the COVID-19 pandemic and so we unfortunately are unable to comment on any changes as a result of the pandemic. We have removed references to the pandemic to avoid confusion. 

3. Briefly highlight the evidence in literature in addition to saying its there, explain it in relation to the findings/study context, include some figures

We have now provided more background literature in the introduction, including specific statistics. 

Highlight the policies on disability that are not being enforced or lacking that will preclude recommendations later on

We do not have any evidence that there were policies that were not being enforced, but this investigation was outside the scope of the study. 

4. In introduction give a systematic regional and global picture for context that will be used as a preamble to the explanations in the discussion

We have added more references for context but in the spirit of being concise have not gone into elaborate detail on a regional/global picture as this would be beyond the scope of the study. 

5. Highlight the problem with access and disability clearly as Maldives has made strides and accommodates various health problems via insurance.

We have now been more specific throughout the paper, particularly in the discussion and conclusion. 

6. Explain the problem using means such as OOP, CHE. Consider reporting mortality or hospital admissions due to chronic illness as opposed to merely reporting a chronic health condition

We do not have data on mortality or hospital admissions due to chronic illness. 

7. Consider adding more exclusion criteria such as severe disability

The definition of disability used in this study is a standard measure and so excluding people with severe disabilities would not be in line with the recommended use of this tool. 

8. The model may be overfitted and may be difficult to replicate. Having too many variables can cause multicollinearity for example, expenditure in last month and expenditure in last 12 months overlap, consider using one of the twoefine what a serious health event is for the problem to be highlighted

We have now clarified how we have run the logistic regression models (see “data analysis”). All outcome variables are not put into the same model, but rather each outcome is compared between people with and without disabilities, with adjustment for age, sex and location. 

9. Consider mentioning how variables affect health status, QALYs for example for readers to appreciate the problem more

We unfortunately do not have this data. 

10. The variable self health rating needs further clarification as one may see it as something any individual with a chronic illness would rate as poor. 

More details have been added under “Selection of indicators”. The purpose of the tools is to document the (perceived) level of health amongst people with and without disabilities, and does not take into account what might be causing differences in perception of health. The utility is more for demonstrating that people with disabilities experience worse health and then other indicators can be used to assess why. 

11. The majority of people with disability in the study sample are aged 40-64+. This is an interesting finding as it may justify the good healthcare system in Maldives and brings in the possibility of occupational hazards being poorly managed and the need for geriatric care to include disability.

The increase in prevalence by age is in line with the global evidence, as ageing is the most common cause of disability (see for example World Report on Disability, 2011). This study did not explore cause of disability in detail, but it may be another area for future research. 

12. Compare episodic costs for disabled and non-disabled to appreciate the high cost, or compare the episodic costs to those experienced by disabled people in countries in the region.

We unfortunately do not have this data for this study. We have added some additional references about catastrophic costs amongst people with disabilities in other countries in the introduction. 

13. Explain the policy or benefits package deficiencies in relation to results for example more effort should be placed in registration, is the study advocating for specific benefits packages for the disabled?

We have now added text on areas for improvement in the discussion and conclusion. 

14. Explain all statements empirically for example good coverage conceals variation in coverage by sub-groups.

We have rephrased this sentence. The underlying data is in the supplemental file. 

15. When comparing results from other studies, highlight the differences or similarities to your study and the meanings they bring.

We have now clarified in the discussion if other studies’ findings are reflective or in contrast to this study’s. 

16. When explaining or stating results, state the significance of the findings in relation to UHC, and recommendations

We have now revised the conclusion to focus more specifically on recommendations. 

17. Consider categorizing variables or findings into knowledge and perception, expenditure/costs, service provision, health status, access (distance, transport) for a systematic flow of the discussion

We unfortunately do not have data on all these categories but we hope the edits to the discussion have now helped with flow. 

18. Revise some grammatical and referencing errors

We have reviewed the manuscript and made some changes but please do flag if you see others. 

Reviewer #2: 

19. LMIC the name is not described before the acronym

Thank you, we have now updated this. 

20. The methodology should be clear about which variables were used in the logistic regression,

We have now clarified the variables used in the logistic regressions. 

21. In children younger than 18 which questions were used to assess disability?

For children 2-17, the UNICEF-Washington Group Child Functioning Modules (sets for children 2-4, 5-17) were used (see second paragraph of “Sample selection”).

22. It will be important to present the average household income of persons with and without disabilities in the sample.

We have now included median household income in Table 1. 

23. It will be recommended to include in the methodology which variables were used to adjust the logistic regresion (apart from sex and age), also to explain what happened to the control variables and to include interactions, such as sex and disability and analyse if the coefficients changes.

We have now expanded on the construction of the regression models in the section “Data analysis”. We are unfortunately unable to explore in detail the effect of gender or other variables with the exception of on unmet need for services, given the sample size, but have added this as an area for further research. 

24. In addition, to include clear policy recommendations in the context of Maldives and to explain why the Maldives is a good country to study this topic, and how other countries can learn from these findings.

We have added these details primarily to the conclusion as well as other places within the discussion. 

From the Editor:

We have updated the manuscript in line with these style requirements. 

26. In the ethics statement in the Methods, you have specified that verbal consent was obtained. Please provide additional details regarding how this consent was documented and witnessed, and state whether this was approved by the IRB. In addition, in your Methods section, please ensure you have also stated whether you obtained consent from parents or guardians of the minors included in the study or whether the research ethics committee or IRB specifically waived the need for their consent.

Further details have been added under “Ethical considerations”. 

27. Please include a complete copy of PLOS’ questionnaire on inclusivity in global research in your revised manuscript. Our policy for research in this area aims to improve transparency in the reporting of research performed outside of researchers’ own country or community. The policy applies to researchers who have travelled to a different country to conduct research, research with Indigenous populations or their lands, and research on cultural artefacts. The questionnaire can also be requested at the journal’s discretion for any other submissions, even if these conditions are not met. Please find more information on the policy and a link to download a blank copy of the questionnaire here: https://journals.plos.org/plosone/s/best-practices-in-research-reporting. Please upload a completed version of your questionnaire as Supporting Information when you resubmit your manuscript.

We have now uploaded this questionnaire. 

28. We note that you have indicated that data from this study are available upon request. PLOS only allows data to be available upon request if there are legal or ethical restrictions on sharing data publicly. For more information on unacceptable data access restrictions, please see http://journals.plos.org/plosone/s/data-availability#loc-unacceptable-data-access-restrictions. 

The data is available on a public repository, so it will not be up to any of the authors to determine who can and cannot access. Any individual wishing to access the data for reproduction or other scientific purposes can submit a request to the repository (https://doi.org/10.17037/DATA.00001698). The data is not fully available as availability upon request was determine most in line with what participants’ had agreed to during the consent process. 

29. In your revised cover letter, please address the following prompts [on data availability]

Data is available in the LSHTM Data Compass repository (https://doi.org/10.17037/DATA.00001698). Users must submit a request to access data, to prevent the data from being used for purposes other than replication or other scientific inquiry. The restrictions were put in place at the request of the data protection office at the grant holding institution. 

30. Please include captions for your Supporting Information files at the end of your manuscript, and update any in-text citations to match accordingly. Please see our Supporting Information guidelines for more information: http://journals.plos.org/plosone/s/supporting-information. 

Captions have been added for the supporting information files and the in-text citations have been updated.

---

## [Decision Letter · Decision Letter 1]

15 Nov 2022

Disability and the achievement of Universal Health Coverage in the Maldives

PONE-D-21-33864R1

Dear Dr. Banks,

We’re pleased to inform you that your manuscript has been judged scientifically suitable for publication and will be formally accepted for publication once it meets all outstanding technical requirements.

Kind regards,

Matteo Lippi Bruni, PhD

Academic Editor

PLOS ONE

Additional Editor Comments (optional):

Reviewers' comments:

Reviewer's Responses to Questions

**Comments to the Author**

1. If the authors have adequately addressed your comments raised in a previous round of review and you feel that this manuscript is now acceptable for publication, you may indicate that here to bypass the “Comments to the Author” section, enter your conflict of interest statement in the “Confidential to Editor” section, and submit your "Accept" recommendation.

Reviewer #1: All comments have been addressed

Reviewer #2: All comments have been addressed

2. Is the manuscript technically sound, and do the data support the conclusions?

Reviewer #1: Yes

Reviewer #2: Yes

3. Has the statistical analysis been performed appropriately and rigorously? 

Reviewer #1: Yes

Reviewer #2: Yes

4. Have the authors made all data underlying the findings in their manuscript fully available?

Reviewer #1: (No Response)

Reviewer #2: Yes

5. Is the manuscript presented in an intelligible fashion and written in standard English?

Reviewer #1: Yes

Reviewer #2: Yes

6. Review Comments to the Author

Reviewer #1: (No Response)

Reviewer #2: it will be advisable to explain in mode detail the analysis of the data and present the justification of the regression model and all the variables

7. PLOS authors have the option to publish the peer review history of their article (what does this mean?). If published, this will include your full peer review and any attached files.

Reviewer #1: No

Reviewer #2: No

---

## [Editor Report · Acceptance letter]

12 Dec 2022

PONE-D-21-33864R1 

Disability and the achievement of Universal Health Coverage in the Maldives 

Dear Dr. Banks:

I'm pleased to inform you that your manuscript has been deemed suitable for publication in PLOS ONE. Congratulations! Your manuscript is now with our production department. 

Kind regards, 

on behalf of

Dr. Matteo Lippi Bruni 

Academic Editor

PLOS ONE